# Lifestyle advice to cancer survivors: a qualitative study on the perspectives of health professionals

Dimitrios A Koutoukidis,[1,2] Sonia Lopes,[1] Abigail Fisher,[1] Kate Williams,[1] Helen Croker,[1] Rebecca J Beeken[1,3]

DAK and SL contributed equally.

[1]Department of Behavioural Science and Health, University College London, London, UK
[2]Nuffield Department of Primary Care Health Sciences, University of Oxford, Oxford, UK
[3]Leeds Institute of Health Sciences, University of Leeds, Leeds, UK

**Correspondence to**
Dr Rebecca J Beeken;
r.beeken@leeds.ac.uk

## ABSTRACT

**Objectives** Adoption of healthy lifestyle behaviours has shown promising effectiveness in reducing the high morbidity burden of cancer survivors. Health professionals (HPs) are well suited to provide lifestyle advice but few survivors report receiving guidance from them. This study aimed to explore HPs' perspective of lifestyle advice (on healthy eating, physical activity, smoking, and alcohol) for cancer survivors.

**Design** In-depth semistructured qualitative interviews were conducted by telephone or face to face. Data were analysed using qualitative content analysis.

**Setting and participants** Twenty-one UK HPs working in secondary care with breast, prostate or colorectal cancer survivors were interviewed.

**Results** The overarching theme was that HPs' desire to provide lifestyle advice was not necessarily matched by knowledge and action. Three main themes were identified: (1) survivorship-centred barriers to provision, (2) HP-centred barriers to provision, and (3) optimal delivery of lifestyle advice. Results suggested that HPs' perceptions of survivors' current status of practising health behaviours, their perceived socioeconomic barriers and ability to practise health behaviours, and HPs' fear for potential loss of connection with the patient influenced provision of lifestyle advice. Further factors included HPs' knowledge of healthy lifestyle guidelines, feeling that they were not the 'right person' to provide advice, and lack of time and resources. HPs stressed that the optimal delivery of lifestyle advice should (1) be tailored to the individual and delivered throughout the cancer journey, (2) be focused on small and achievable changes framed as part of their treatment regimen and (3) be cost-effective for wide-scale implementation.

**Conclusions** Incorporation of the identified barriers when developing HP training programmes and lifestyle interventions could increase the probability of successful behavioural change, and thus improve outcomes for cancer survivors.

## INTRODUCTION

There are 2.5 million cancer survivors living in the UK.[1] The increasing number of cancer survivors and their higher morbidity burden compared with the general population means strategies are needed to improve survivorship outcomes.[2 3] Adoption of key modifiable

### Strengths and limitations of this study

► The study comprehensively examined the views of health professionals, working with breast, prostate and colorectal cancer survivors, on key modifiable lifestyle behaviours.
► Health professionals had diverse professional backgrounds, including nursing, oncology, surgery and allied health professions.
► It is possible that the participants were generally interested in lifestyle topics, which might bias the responses towards a positive view on lifestyle.
► Although qualitative studies provide in-depth information, their findings cannot be generalised.

lifestyle behaviours (ie, not smoking or drinking alcohol, eating a healthy diet, engaging in sufficient physical activity and weight management) are potential strategies. There is a growing body of evidence that suggests these health behaviours are positively associated with improved outcomes in cancer survivors.[4 5] Furthermore, behavioural lifestyle interventions in this population have shown promising effectiveness.[6 7]

There are now lifestyle guidelines for cancer survivors, advocating smoking cessation, moderate alcohol consumption (if any), a healthy balanced diet, daily physical activity and maintenance of a healthy weight.[8–11] Although surveys suggest that few cancer survivors are meeting these guidelines,[12] qualitative data suggest that a cancer diagnosis may promote motivation for behavioural change and that survivors are keen to receive healthy lifestyle advice, ideally from a credible source.[13 14] Health professionals (HPs) are well positioned to provide healthy lifestyle advice and, thus, help cancer survivors improve their lifestyle behaviours. However, few survivors report receiving lifestyle advice from their HPs.[13 15]

We previously conducted a survey with HPs suggesting that 36% of HPs were not aware

of any lifestyle guidelines for cancer survivors.[16] Although most reported providing some lifestyle advice, a number of barriers to provision were also cited, including lack of guidelines, belief that advice would not change behaviour or affect outcomes, belief that they are not the right person to provide advice, lack of time, lack of patient interest, fear that they would offend the patient and cultural differences. These highlighted the potential for an intervention to help HPs provide evidence-based advice.

The aim of this qualitative study was to explore in greater depth HPs' views on the provision of lifestyle advice to patients with a cancer diagnosis.

## METHODS
### Procedure and participants

Eligible participants were UK-based HPs (including surgeons, physicians, nurses and allied HPs) caring for patients diagnosed with the most common lifestyle-related cancers (breast, prostate or colorectal cancer). They had provided contact details following their participation in an online survey on lifestyle guidelines for cancer survivors and agreed to be contacted for future research.[16] HPs were invited via email to participate in either a telephone (n=20) or an individual face-to-face (n=1) interview at University College London (UCL). Participants provided signed informed consent.

### Interviews

Between October 2014 and January 2015, an experienced female research psychologist with interest in cancer survivorship (SL) interviewed the HPs. The interviewer had no previous relationship with the study participants. Participants were aware that the study was part of the interviewer's research degree. Each interview lasted approximately 40 min (range: 17–74 min). The interview followed a semistructured guide (online supplementary material 1). The guide followed up on HPs' individual survey responses about the provision of advice to patients who had completed treatment, and used open-ended questions and prompts to explore in-depth HPs' awareness of survivorship lifestyle guidelines, their barriers to provision of lifestyle advice and their views on various formats for lifestyle interventions. Interviews were audio-recorded, anonymised and transcribed verbatim. The researcher listened to the tapes to verify them for accuracy. Field notes were not taken as all, but one interview took place over the phone, meaning information could not be collected on setting, appearance or non-verbal behaviours of participants.

### Analysis

Interview transcripts were analysed using qualitative content analysis[17] in NVivo V.10. Content analysis was chosen as the main aim of the study was to describe HPs views as opposed to developing theory or exploring the lived experience.[18] Following familiarisation with the data, meaning units and codes were generated using an inductive approach. SL

coded the transcripts derived the coding list (online supplementary material 2) in an iterative process with RJB. An independent reviewer with a dietetics background (DAK) coded 20% of interviews against the coding list. Inter-rater reliability was high (mean kappa=0.92) and minor coding differences were resolved by discussion. Categories and themes were compiled with a manifest analysis using an iterative process. This ensured internal homogeneity and external heterogeneity of the themes and categories. The completed consolidated criteria for reporting qualitative research checklist is available in online supplementary material 3.[19]

## RESULTS
### Sample characteristics

Fifty-two of the 336 survey participants were eligible for this study. Twenty-one (40.4%) were interviewed and 31 either did not respond or were unwilling to take part. Table 1 shows the sample characteristics of the study in relation to the survey. Most participants were female and aged between 46 and 55 years. Their professional role varied and their work location spanned across the UK.

### Themes

The overarching theme arising from the interviews was that HPs' desire to provide lifestyle advice was not necessarily matched by knowledge and action. Three main themes were identified: (1) HP's perceptions of survivor-centred barriers to provision of lifestyle advice, (2) HP-centred barriers to provision of lifestyle advice and (3) optimal delivery of lifestyle advice.

#### HPs' perceptions of survivor-centred barriers to provision of lifestyle advice
*Perceptions of survivors' current health behaviours*

HPs felt that lifestyle behavioural change was particularly challenging for cancer survivors, due to ingrained life-long habits and sustainability of achieved changes.

> Well, lifestyle is very much a matter of habit, and breaking habits is always difficult for all of us. Breast Cancer Surgeon, M, 63y

Coexistence of multiple suboptimal behaviours was regarded as a barrier to provision of lifestyle advice.

> It is sometimes difficult when people just do seem to have all the vices, so they are heavy smokers, they are not just overweight, they are really obese, they obviously take no exercise and they are heavy smokers. It's very difficult to know sometimes where to start. Breast Cancer Surgeon, F, 53y

Alcohol and tobacco were regarded as sensitive topics and it was perceived hard to assess consumption levels. Therefore, this was only addressed if there were evident indications of a problem or if the topic was raised by relatives.

> […] when it comes to alcohol it can be quite tricky. Sometimes we'll have relatives who will say, 'Are you

**Table 1** Sample characteristics in relation to the previous survey

| | Interview participants, n=21 | Survey participants, n=336* |
|---|---|---|
| | % (n) | % (n) |
| **Gender** | | |
| Female | 76 (16) | 81 (272) |
| Male | 24 (5) | 19 (64) |
| **Age (in years)** | | |
| 26–35 | 5 (1) | 12 (41) |
| 36–45 | 38 (8) | 33 (110) |
| 46–55 | 48 (10) | 42 (141) |
| 56–65 | 10 (2) | 13 (42) |
| **Profession** | | |
| Nurse | 43 (9) | 55 (126) |
| Physician | 33 (7) | 21 (48) |
| Surgeon | 19 (4) | 13 (31) |
| Allied HP (physiotherapist) | 5 (1) | 11 (26) |
| **Cancer specialism** | | |
| Breast | 43 (9) | 25 (54) |
| Colorectal | 38 (8) | 15 (32) |
| Prostate | 19 (4) | 10 (21) |
| **Region of work** | | |
| North West England | 38 (8) | – |
| London | 24 (5) | – |
| South West England | 10 (2) | – |
| South East England | 10 (2) | – |
| Wales | 10 (2) | – |
| East of England | 5 (1) | – |
| Scotland | 5 (1) | – |

Percentage may not total 100 % due to rounding.
*Data from Williams *et al*[16].
HP, health professional.

aware this person is drinking?' particularly if they are on chemotherapy. And the side effects of chemotherapy, and then to add alcohol onto it. I think that's a more difficult area to try and challenge people on. Colorectal Cancer Nurse, F, 54y

They felt that low mood and distress were also barriers for implementation of the provided advice.

If someone is depressed or has low mood, for whatever reason, everything is harder. Breast Cancer Nurse, F, 40y

### *HPs' perception of survivors' ability to perform health behaviours*
The lifestyle advice provided was mostly tailored to the individual patient. Frailty and physical inability to perform physical activity significantly hindered the HPs'

inclination to provide physical activity advice. Tolerability of a high-fibre diet due to treatment-related bowel changes was also considered a barrier for provision of advice on healthy eating.

If somebody is frail and unwell generally, either because of their cancer or because of other medical problems, then I think advising them about diet and exercise and things isn't necessarily helpful for them. Prostate Cancer Physician, F, 34y

I think some of our patients have got possibly some degree of stenosis, and actually saying to them to have a lot of fruit and fibre in their diet is wrong information. So, I think it has to be tailored to the individual, really. Colorectal Cancer Nurse, F, 54y

### *Potential loss of connection with the patient*
Fear of blaming the patient resulted in HPs taking a subtle rather than a strong approach to providing lifestyle advice. HPs did not want to make people feel guilty about their previous lifestyle or even create future blame if the patient did not subsequently change their lifestyle and the cancer reoccurred. They mentioned a potential loss of credibility and connection with their patients if they were to insist on lifestyle change.

I need to keep credibility with them as well, and if I start making demands of them which they really can't possibly keep then I kind of lose my own connection with these people. Breast Cancer Surgeon, M, 63y

Another barrier was the obstruction of patients' free will, as they believed people have their own views about what is healthy or unhealthy. Some felt that they should only start such a conversation if the patients were to bring up the topic. HPs perceived lifestyle to be a low priority for the majority of survivors, although they acknowledged the cancer diagnosis as a 'teachable moment' for them.

You have to be realistic as well. People will always make the wrong choices or choose to drink too much alcohol, or chose to smoke, or whatever, and that's freewill, isn't it, as well? Breast Cancer Nurse, F, 46y

They don't see it as a priority. Their priority if much more about "What's happening with my cancer? Has it returned?". The financial issues, visa issues, all that sort of stuff. So, diet and lifestyle is somewhat down the line for a lot of patients. Colorectal Cancer Nurse, F, 55y

I mean, a smoker, if they are determined not to give up there's absolutely nothing you can do about it. I mean, for some it's a real wake-up call and for some they really change their lifestyle. […] and then also sometimes people don't want to talk about it. Prostate Cancer Physician, F, 54y

### *Socioeconomic barriers to practising health behaviours*
HPs perceived practical barriers for survivors included affordability of a healthy diet or gym subscription.

Furthermore, perceived lack of social support and cultural variation were also reported as a barrier to practising health behaviours.

> I think the difficulty is that depending on the demographics of patients you've got how practical it can be to either give the information or encourage people to follow the guidelines. As I say, our demographics are such that there's a lot of poverty within our catchment area. So that's often the biggest barrier to doing that. There's also… depending on cultural backgrounds, some of our patients are not keen to take responsibility, they prefer the paternalistic approach. So that can be quite difficult. We also have a lot of patients who are very socially isolated, and are not inclined to go shopping, or cook for themselves. Colorectal Cancer Nurse, F, 55y

### HP-centred barriers to provision of lifestyle advice
*Knowledge and attitudes towards evidence and guidelines for health behaviours*
HPs tended to be aware only of general lifestyle guidelines, such as the Department of Health alcohol guidelines, rather than specific guidelines for cancer survivors. Others were not aware of any guidelines.

> I've come across research papers about things like that [smoking and diet], but I haven't come across any guidelines. Physiotherapist, F, 51y

Their attitude towards these general guidelines was positive, but they based their advice only loosely on these. The advice tended to focus on either general health or controlling side effects and improving recovery. Only some HPs advised survivors that a healthy lifestyle might improve disease outcomes.

> I do take the guidance into consideration but I do a bit more patient-centred approach, really. Colorectal Cancer Nurse, F, 51y

> So, I have always preached to patients that it's actually good to go our for a walk, get some fresh air, that sort of thing. But, obviously, more recently, there's been data both… for many of the polyp sort of cancers—whether it's breast or urological cancers—that getting physical exercise does improve outcomes. So, from the historical data to much more robust data, I'm pretty sold on the idea that just getting fresh air, going out for a walk, sensible exercise, not being… not heroic exercise, is good for the mind, good for the body, and good for physical recovery. Prostate Cancer Physician, M, 49y

They also regarded the evidence on healthy lifestyles, cancer prevention and cancer recurrence as not definitive. They perceived the evidence for physical activity on cancer recurrence to be stronger than that available for any dietary factors; however, they were still advocates of a healthy diet for overall health. There were mixed views regarding the role of lifestyle in cancer recurrence, with only some professionals aware of this link and research findings.

> Well, I suppose that it helps to reduce the risk of recurrence. A good healthy diet and to take a good… you know, exercise. Breast Cancer Physician, M, 40y

> I wouldn't say that diet and physical activity are related to cancer recurrence. Colorectal Cancer Physician, F, 41y

The fact that lifestyle is one of multiple cancer risk factors made them less inclined to focus their consultations on it.

> There's obviously other factors at work and at play and, although I encourage patients to take exercise and to eat a healthy diet, particularly with lots of veg, I don't want to be creating a guilt feeling that, if they do all of this, it will stop it coming back. And therefore, if it comes back it's because they didn't eat the right diet. So I think it is important, but there's so many other factors involved that I don't want to concentrate on this to the detriment of others. Colorectal Cancer Physician, F, 44y

In contrast, evidence to back claims about the impact of behavioural change was seen as both important to facilitate change and as something that patients do not pay attention to.

> Perhaps it would be helpful for patients to have some information that specifically says: 'We know that patients who stop smoking… We know that patients who are more active… We know that patients who keep within a healthy weight…' and then it has those percentages for reduced risk of recurrence, etc. […] perhaps it would motivate them. Breast Cancer Nurse, F, 46y

HPs also mentioned that they could not quantify the effect of lifestyle changes post-treatment in terms of any outcome, given the lack of research data. This made their argument weak and the conversation more challenging than those conversations focusing on medical treatments.

> I can tell them if you take endocrine therapy, for survival, it would be improved by 10%. Chemotherapy, another 10%. So, 20% improvement. So there is some evidence for them for me to talk to the patient with the other treatments I'm offering, but I don't have anything for these [healthy lifestyle]. So that's another reason I won't be able to give them full information, that's when I say lack of information to give to the patients. Breast Cancer Surgeon, M, 44y

*Self-identification as the right person to provide lifestyle advice*
While some felt it was their duty to empower survivors to lead a healthier lifestyle, others thought that they were not the right person to provide lifestyle advice, given their limited knowledge on the topic and other priorities during their short consultations.

And I think we have a duty to advise patients on quitting smoking and reducing alcohol intake and stuff. Colorectal Cancer Nurse, F, 38y

I don't think it's necessarily my role to provide generic lifestyle advice to everybody, and unfortunately there isn't the time. I tend to focus on the people who have a problem, like fatigue, which I think would respond to exercise. Prostate Cancer Physician, F, 34y

Some HPs thought their authority might help survivors take the advice more seriously, while others perceived their advice to be mostly ineffective.

I don't think that doctors necessarily are the best [to provide advice] unless they are felt to be a, sort of, place of authority for the patient. Prostate Cancer Physician, F, 54y

There aren't many people I've advised to stop smoking who have ever done it. So I'm fairly sceptical about the value of my advice even though I persist with it. Breast Cancer Physician, M, 64y

HPs perceived their own lack of adherence to lifestyle guidelines as both a barrier and a facilitator for providing advice; by not being a role model and by being able to relate to the challenge of behavioural change during the conversation with patients, respectively.

I don't have a BMI under 25, so I have no room to talk. Sometimes, though, a bit of 'I know how you feel' and a bit of feeling goes a long way. I know that one of my breast cancer nurses, who is very slim and very athletic, finds it far more difficult to talk to people about losing weight, because she doesn't feel she has anywhere to come from. Breast Cancer Surgeon, F, 53y

*Practical barriers to lifestyle advice provision*
HPs regarded lack of time as a significant barrier for provision of lifestyle advice.

I don't have enough time to do it all properly and sensitively. Colorectal Cancer Physician, F, 44y

They mentioned that they mostly provide verbal lifestyle advice due to a lack of other type of resources. Some tended to reiterate this advice in their patient letters, provide leaflets generated by their local multidisciplinary team or refer people to existing relevant services for further support.

If some patients I think will benefit from a proper programme, then we give them this Living Well leaflet. If some patients who are saying they only have exercise and they are happy with it, then we don't given them that Living Well programme or healthy lifestyle leaflet. Apart from that, I don't have any other leaflets about diet or exercise as such, a specific how it improves their breast cancer survival. So in those questions I don't have nothing much to give them. And in my letter, like one line, I'll just mention to the patient the importance of maintaining a healthy weight, diet, and exercise. Breast Cancer Surgeon, M, 44y

### Optimal delivery of lifestyle advice
*Tailored advice delivered throughout the cancer journey*
HPs stressed that there is not a one-size-fit-all optimal format of lifestyle advice and were keen on providing choice of multiple formats, so that the patient could choose the one that fit best their individual needs and preferences. This was thought to enhance intervention adherence.

The level of intervention will depend on the patients, their background, the sort of feedback they give us to their motivation. Prostate Cancer Physician, M, 49y

I think the best format is that somebody talks to the patient and asks them what they would want, and then plan the intervention accordingly rather than having one fitting for everybody, because that's not going to help. Prostate Cancer Physician, F, 54y

The views on the optimal timing of providing advice were varied with some suggesting to intervene at diagnosis and others at the end of treatment. Some suggested that the optimal timing is patient dependent. Others felt that the advice should be initiated at diagnosis and reiterated in each follow-up appointment.

It's something that should have been on the agenda all the way through and then [at the end of treatment] you're literally summarising, concluding, and then exploring if there's something they want to take over to another level. Prostate Cancer Physician, M, 49y

HPs mentioned that a combination of verbal and written lifestyle information would be preferable to only verbal advice, as patients receive too much information during their cancer journey and they cannot take in everything. This was viewed as preferable to written advice alone as it could be made more relevant to the patient. HPs felt that all HPs should provide lifestyle advice.

If I just gave them a leaflet… I think it needs to be given in conjunction with a health professional or somebody who can talk them through it. If you just give them another leaflet, they're just likely to put it with their pack of other things they've got. Physiotherapist, F, 51y

HPs in favour of advice using a face-to-face meeting with a HP felt that this meeting should be part of the general follow-up to boost attendance and acceptability. They mentioned that the meeting could enhance continuity of care post-treatment, fostering personal contact with a professional aware of the patient's journey. For the survivors who would prefer the support of a group-based intervention, local community venues were preferred

over hospital rooms. Online advice was deemed appropriate for technology-literate survivors.

> A group intervention is definitely a positive thing for patients to get that peer support from other patients as well. I think it helps patients to feel that they can manage their condition better themselves and also with support from other patients rather than it just being medically led all the time, particularly if it's off site as well, if it's in other, perhaps, community settings as well. Breast Cancer Nurse, F, 46y

> I think online is probably the best thing, if patients have access and they're computer literate, because they can do it at their own time and at their own pace. Colorectal Cancer Nurse, F, 55y

### Small and achievable changes framed as part of their treatment regimen

HPs mentioned that survivors who need to change multiple behaviours might be overwhelmed by the process and suggested prioritising and tackling one or two behaviours at a time focusing on small realistic changes.

> I'm aware that if you give people too much advice all at once, it's quite difficult for them to take it all in, so sometimes you are better off just focusing on one thing [...] finding out which you think is the most risky of their behaviours and concentrate on that one. Prostate Cancer Nurse, F, 53y

Some thought that framing the healthy lifestyle advice as part of their treatment would make it more engaging.

> It would be nice to have some guidelines available, some specific booklet available about benefits of healthy diet and exercise, so we can give it to them as well, so they can follow it as a specific treatment regimen. [...] It is a treatment – healthy diet and exercise – and say it's a treatment. We should make them feel like that. Breast Cancer Surgeon, M, 44y

> I'd start telling them from the moment I see them with their diagnosis, because I want them to feel it's as important for them to undertake physical activity and eat a good diet and lose some weight as is their other treatments, like radiotherapy, and hormone blocking treatment. [...] I tell them it will be part of their treatment so that they are prepared and accept it as normal. Breast Cancer Surgeon, F, 41y

### Cost-effectiveness for wide-scale implementation

Time and financial resources and proven cost-effectiveness were cited as barriers for wide-scale implementation of lifestyle advice. Lower need for resources indicated higher potential for implementation, but HPs were unsure of measurable outcomes for one-off low-intensity lifestyle interventions, such as a leaflet or a phone call.

> I think the patients would like the face-to-face. I suppose it just depends on what the resources are, as to who is doing it – is it a dietitian, is it a CNS, is it another organisation? So I think it depends on the manpower, what resources are available for a face-to-face. I think if it was extra on top of what we currently provide then we would struggle. Colorectal Cancer Nurse, F, 54y

> Integration is fine [for leaflet and online advice]. The question is: will it help? Prostate Cancer Physician, F, 54y

## DISCUSSION

The current study reiterated that HPs' desire to provide lifestyle advice is not necessarily matched by knowledge and action. Reported barriers to provision included perceptions about survivors' behaviours, abilities and interest in advice, as well as concerns about HPs' own abilities to provide effective advice without damaging their patient relationships. Nonetheless, HPs in this study were keen for most survivors to receive lifestyle advice throughout the cancer journey. Suggestions for this advice included tailoring to individual patient needs and preferences, initiating small and achievable change goals and framing recommendations as part of the patient's treatment regimen.

Previously reported barriers to provision of lifestyle advice include perception that patients lack interest in behavioural change, perception that behavioural change is not feasible or effective, lack of guidelines and confusion about the research findings, self-identification as not the right person to provide advice, potential for blaming the patient, lack of appropriate behavioural change support and lack of time.[16 20 21] The current study confirmed the above barriers offering further insight, such as the perception that providing advice on alcohol and smoking is more challenging compared with diet and physical activity. Another insight was that the fear of blaming relates to cancer onset and to cancer recurrence. The study also added to the list of barriers the potential loss of credibility and connection with survivors, the obstruction of free will, the importance of role modelling and the challenge of addressing the coexistence of multiple suboptimal behaviours given practical barriers to implementing advice, such as affordability of a healthy diet. These factors could be taken into consideration in the development of programmes aiming to train HPs to effectively deliver lifestyle advice, as evidence suggests that even brief advice from HPs is effective for lifestyle behavioural change.[22–24]

Interestingly, the HPs' self-identification as not the right person to provide advice and their fear of losing the connection with their patients by providing them with continuous lifestyle advice, are not matched by the fact that survivors would actually welcome lifestyle advice from their oncology HPs.[13–15] Survivors' views partly support the perception of HPs that it is difficult to change engrained unhealthy life-long habits. However, although there is little evidence that survivors make major changes in their health behaviours following a cancer diagnosis,[8 12] some report motivation

to engage in behavioural change.[13 25] HPs, therefore, have an opportunity to capitalise on this and to provide lifestyle advice to survivors or signpost them to further support. These interventions should include social support, because this behavioural change technique was perceived to facilitate performing health behaviours by both HPs in the current study and survivors.[15]

However, HPs in this study did not mention reputable and long-standing lifestyle guidelines for cancer survivors.[11] One reason for the lack of awareness might be that these guidelines are based on the cancer prevention literature. However, the evidence base for their implementation post-diagnosis is growing, and they are considered to be prudent advice for cancer survivors. The lack of dissemination of these guidelines should therefore be targeted with educational courses, conferences and training to enable HPs to provide evidence-based advice. Professional organisations might be pivotal in this process. Programmes to enhance oncology HPs knowledge on lifestyle guidelines and facilitate provision of advice are currently being piloted, and are showing promising potential.[26 27] These programmes should ensure adequate knowledge of the guidelines and provide the tools and techniques to HPs to ensure they approach the topic of behavioural change in a supportive way with confidence and sensitivity, so that cancer survivors feel motivated and not guilty.

HPs in this study also indicated that the format of lifestyle advice should be tailored to individual survivors. Surveys, qualitative studies and analysis of data from large behavioural change interventions are required to identify factors that predict cancer survivors' uptake and adherence to various types of lifestyle interventions. However, an analysis of three large home-based lifestyle interventions in cancer survivors did not identify any predictors of adherence to the different types of intervention.[28] To that end, quasi-experimental intervention designs (eg, face-to-face vs remote intervention) based on survivors' preferences for delivery format might better facilitate understanding of adherence and lead to effective personalised lifestyle support. Framing the intervention as part of the treatment regimen instead of health promotion advice could also be explored in future studies, with pilot studies indicating the effectiveness of physical activity rehabilitation on fitness levels.[29]

Strengths of the current study include the examination of HPs' views on key modifiable lifestyle behaviours from a sample of various HPs from across the UK. Although qualitative studies provide in-depth information, their findings cannot be generalised. It is possible that the participants of this study were generally interested in lifestyle topics, which might bias the responses towards a positive view on lifestyle. Furthermore, the views may not be representative of all oncology HPs. Further research could explore this, as HPs working with other cancers may be even less likely to give advice.

In conclusion, the findings of this study suggest that some, but not all, HPs specialising in breast, prostate and colorectal cancers provide lifestyle advice to their patients post-treatment. Those who provide advice describe providing generic guidance broadly based on healthy lifestyle recommendations for the general population, with attempts to tailor it to the individual patient. Additional education is needed to improve awareness of available guidelines for cancer survivors. Incorporation of the identified factors that influence the provision of lifestyle advice in the development of training programmes for HPs and lifestyle interventions for cancer survivors could increase the probability of effectiveness and in turn, improve outcomes for cancer survivors.

**Acknowledgements** The researchers are grateful to the health professionals who participated in the study.

**Contributors** RJB conceived the study and obtained the funding for the Impact Award Studentship. SL, RJB, AF, KW and HC devised the interview protocol. SL conducted the interviews. SL, DAK and RJB analysed the data with contribution from all authors. DAK drafted the manuscript. All authors critically revised and approved the final version of the manuscript.

**Funding** All authors were supported by funding from Cancer Research UK. SL was also funded by University College London and the Weight Concern Impact Award Studentship. RJB is currently supported by Yorkshire Cancer Research Academic Fellowship funding.

**Competing interests** None declared.

**Patient consent** Not required.

**Ethics approval** Ethical approval was granted by the UCL Research Ethics Committee (project ID: 4456/001).

**Provenance and peer review** Not commissioned; externally peer reviewed.

**Data sharing statement** Anonymised interview transcripts can be obtained by the corresponding author on reasonable request.

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
