## [Reviewer comments · BMJ Open]

ARTICLE DETAILS

TITLE (PROVISIONAL)	Lifestyle advice to cancer survivors: a qualitative study on the perspectives of health professionals
AUTHORS	Koutoukidis, Dimitrios; Lopes, Sonia; Fisher, Abigail; Williams, Kate; Croker, Helen; Beeken, Rebecca

VERSION 1 – REVIEW

REVIEWER	Claire Foster and Amy Din Faculty of Health Sciences, University of Southampton
REVIEW RETURNED	14-Nov-2017

GENERAL COMMENTS	This was an interesting paper describing a topic of current concern, and provides justification for further research to understand the provision of lifestyle advice to cancer survivors. This manuscript describes a qualitative project involving a relatively small sample (N=21) of health professionals (HPs: working with breast, prostate, colorectal cancer patients) in order to understand their knowledge of cancer and lifestyle guidelines as well as the advice HPs are giving to cancer survivors (or not). Where current knowledge is lacking, this study also aims to understand the provision of lifestyle advice by HPs to survivors of lifestyle-related cancers. This qualitative project was informed by an earlier survey conducted by Williams et al (2015). The approach chosen was a qualitative approach involving semi-structured interviews with 21 HPs and analysing the transcripts using content analysis. Due to the chosen form of analysis the findings are descriptive. This findings provide a useful starting point and may support intervention development to facilitate health professionals provide lifestyle advice to survivors. Prior to publication we feel it is important to address the following to enhance the paper: 1. The aim could benefit from being restructured2. It would be helpful to include justification for why HPs of breast, prostate, or colorectal were selected. This has been provided in line 472, but should be clarified in the first instance3. Clarify eligibility criteria (eg on quote refers to end of life care which is quite different).4. Justify content analysis as chosen methodology5. Barriers to supporting cancer survivors to make lifestyle changes seems to have limited attention in the interview schedule – was this reflected in the transcripts?6. Could you elaborate on why field notes weren't taken?
---

	7. Please clarify what is meant by 'comprehensive examination' of HPs views? 8. In section 1.2 there is a quote that refers to frailty but not for tolerability of a high-fibre diet. Can a quote be provided for this as well? 9. Line 188-191: statement doesn't feel justified by the quotes provided 10. Line 340: Typo 11. Discussion – avoid statements that present as definitive conclusions Line 475-477: The conclusion needs to reflect the results. For example, HPs described barriers to provision of lifestyle advice, such as only having capacity to provide general advice rather than giving tailored information or providing advice that will elicit a survivors to make behaviour change. Should line 476 read "of those who provided advice", as currently it gives the impression that respondents provided advice 12. This year the WCRF also released CUP reports for both breast and colorectal cancer. These could also be included in your references
--	---

REVIEWER	Dr.Helen Crank The Centre for Sport and Exercise Science Sheffield Hallam University United Kingdom
REVIEW RETURNED	29-Nov-2017

GENERAL COMMENTS	Reviewer Helen Crank (SHU)	
	Please note these are suggestions for the authors to consider. Mainly, they are offered to further the ease of reading and understanding of the manuscript.	
	Line number	Comment/ Suggestion
	Abstract	
	L22	semistructured - change to semi-structured
	L28	2) Clinician-centred barriers - are you using the term clinician interchangeably with HP's -? Are you specifying a type of professional by using the term clinician? can you check through-out if you want to keep in clinician or just use HP.
	L29	"Results suggested that survivors current" ...could you pre-face with "HP's perceptions of survivors..." just to make it clear you are reporting from the HP's point of view on survivors behaviours.
L44	"on all major lifestyle topics" - consider revising to on the key modifiable lifestyle behaviours	

	L72	Consider revising to - previously we conducted a survey with HP's suggesting - to make it explicit that work was already undertaken.
	L77	"belief that they would blame the patient" - not easy to know what you mean by this - perhaps remove.
	Methods	Could you clarify whom was invited to take part in the interview study - was it all previous survey respondents received an email invitation? see next point
	Results	
	L119	Fifty-two of 336 survey participants were eligible for this study - was it that they were eligible - i.e. you had eligibility criteria? Or that 52 agreed to take part? I'm asking as Table 1 suggests all survey respondents might be eligible. Just clarify please.
	L131	Title - Survivor-centred barriers to provision... again for clarity might you preface with "HP's perceptions of" survivors centred... just to make it really explicit this was primary research with HP's not cancer survivors.
	L138	sub-optimal (hyphen)
	L155	Title - consider revising to - HP's perception of cancer survivors ability to perform health behaviours. It also seems the first quote is talking about appropriateness of advice that might be given
	L167	mood and distress sentence and quote might sit better in the previous 1.1 section - patient barriers
	L223	I think there are some important distinctions to be made between giving general lifestyle advice with the intention of improving quality of life, general health and advice that is specifically focussed upon reducing risk of disease recurrence. Can you clarify the point you make about the evidence for physical activity being stronger - was that in terms of general health advice or in terms of reducing further cancer recurrence? Do you have any further data that gives any insight as to for what purpose specifically HP's give advice - it is generic healthy lifestyle advice or do they say

		it's for specific reducing further disease risk? Is there any consensus you can glean from your data? I appreciate you have stated that HP's do not know the evidence base sufficiently well to make clear recommendations - but is that what they want to be able to do?
	L256	Clinicians or HP's? - choice of term Please can you clarify sentence - couldn't quantify effect of lifestyle changes post-treatment - in terms of survival, recovery, wellbeing?
	L290	They - can you say HP's - to be explicit who you are referring to
	L342	Clinicians or HP's
	L346	They felt all HP's should provide lifestyle advice - who is they?
	L352	Those in favour - again just clarify who you are referring to
	Discussion	
	L411	achievable goals framed as part of their treatment regimen - I agree this sounds like a good plan - do you think the strength of the presented data/quotes is strong enough to infer this is what HP's wanted / were meaning?- that lifestyle advice and changes be viewed as treatment regimen? I think you would need to substantiate with more quotes to make this bold statement. When you read the quotes it suggests that HP's don't have the knowledge re evidence, have perceptions of patient's behaviours and fear losing that connection - would you just consider how you might be able to substantiate that point further. Maybe the term "care" is more appropriate - treatment has a medical implication?
	L440	Reference 11 - you make a valid point about the cancer prevention recommendations - could you add / refer to the Rock (2012) guidelines specifically for cancer survivors.
	General	Clearly written, good supporting up to date referencing, interesting points that affirm there is still a lot of work to do to incorporate effective lifestyle advice for cancer survivors into routine cancer care.

VERSION 1 – AUTHOR RESPONSE

We would like to thank the editors and reviewers for taking the time to review our manuscript and giving us thoughtful feedback. We have tried to address their comments and hope that this has improved our manuscript. We have also updated the page numbers in the COREQ checklist to reflect these changes. Please see our responses below, and the corrections and additions in the revised manuscript.

Reviewer 1

1. The aim could benefit from being restructured.

Response: Thank you, we have restructured our aim to read: 'The aim of this qualitative study was to explore in greater depth HPs' views on the provision of lifestyle advice to patients with a cancer diagnosis' (page 4, lines 78-79).

2. It would be helpful to include justification for why HPs of breast, prostate, or colorectal were selected. This has been provided in line 472, but should be clarified in the first instance

Response: As suggested, we have moved this justification to the beginning of the methodology section (page 5, lines 82-84).

3. Clarify eligibility criteria (eg on quote refers to end of life care which is quite different).

Response: We apologise that this was unclear and agree that end of life care is quite different. We asked HPs about the lifestyle advice they provided to survivors focusing on the post-treatment period. However, HPs did mention that they would find it inappropriate to provide advice at the end of life stage. Therefore, we thought that it would be useful to report this. However, we agree it is confusing and have removed this statement. We have also added to our methodology that the interviews were about the provision of advice to patients who had completed treatment (page 5, line 98-99).

4. Justify content analysis as chosen methodology

Response: We have now added the following detail to our methodology 'Content analysis was chosen as the main aim of the study to describe HPs views as opposed to developing theory or exploring the lived experience.' We wished to employ a relatively low level of interpretation and so content analysis was appropriate as opposed to those approaches that require a higher level of interpretive complexity (page 5, lines 108-109).

5. Barriers to supporting cancer survivors to make lifestyle changes seems to have limited attention in the interview schedule – was this reflected in the transcripts?

Response: We are unclear about this point, as topic 3 of the interview schedule covers barriers to provision of lifestyle advice with 9 questions. Every participant discussed the barriers in detail and this was reflected in the transcripts and results.

6. Could you elaborate on why field notes weren't taken?

Response: Field notes were not taken as all but one interview took place over the phone, meaning information could not be collected on setting, appearance or nonverbal behaviours of participants. We have added this information to our methodology (page 5, line 103-105).

7. Please clarify what is meant by 'comprehensive examination' of HPs views?

Response: We apologise that this was unclear. We meant that we covered the key modifiable health behaviours. This has now been corrected.

8. In section 1.2 there is a quote that refers to frailty but not for tolerability of a high-fibre diet. Can a quote be provided for this as well?

Response: We have now provided a quote for this as well (page 8, lines 173-176).

9. Line 188-191: statement doesn't feel justified by the quotes provided

Response: Further quotes have been added which we hope better justify this statement (page 9, lines 194-202).

10. Line 340: Typo

Response: Thank you, the typo has been corrected.

11. Discussion – avoid statements that present as definitive conclusions

Response: Apologies, we have removed these statements and presented them more cautiously.

12. Line 475-477: The conclusion needs to reflect the results. For example, HPs described barriers to provision of lifestyle advice, such as only having capacity to provide general advice rather than giving tailored information or providing advice that will elicit a survivors to make behaviour change. Should line 476 read “of those who provided advice”, as currently it gives the impression that respondents provided advice

Response: Thank you, we have now reworded this sentence in line with this suggestion (page 18, line 505).

13. This year the WCRF also released CUP reports for both breast and colorectal cancer. These could also be included in your references

Response: Thank you for these suggested references. While they are of interest, we have chosen not to include these given their focus on prevention. We feel the more generic WCRF guidelines, while older, are more relevant as they specifically address survivorship.

Reviewer 2

1. L22 semistructured - change to semi-structured

Response: Thank you, this has now been corrected.

2. L28 2) Clinician-centred barriers - are you using the term clinician interchangeably with HP's -? Are you specifying a type of professional by using the term clinician? can you check through-out if you want to keep in clinician or just use HP.

Response: This has now been corrected and “HPs” is used throughout the manuscript.

3. L29 "Results suggested that survivors current". could you pre-face with "HP's perceptions of survivors." just to make it clear you are reporting from the HP's point of view on survivors behaviours.

Response: Thank you, this has now been corrected (page 2, line 28-29).

4. L44 "on all major lifestyle topics" - consider revising to on the key modifiable lifestyle behaviours

Response: We have revised this as suggested (page 3, line 44).

5. L72 Consider revising to - previously we conducted a survey with HP's suggesting - to make it explicit that work was already undertaken.

Response: We have revised this as suggested (page 4, line 70).

6. L77 "belief that they would blame the patient" - not easy to know what you mean by this - perhaps remove.

Response: As suggested, we have removed this statement.

7. Methods Could you clarify whom was invited to take part in the interview study - was it all previous survey respondents received an email invitation? see next point

Response: We apologise that this was unclear. Eligible participants were UK-based HPs (including surgeons, physicians, nurses and allied HPs) caring for patients diagnosed with the most common lifestyle-related cancers (breast, prostate, or colorectal cancer), who had provided contact details following their participation in an online survey on lifestyle guidelines for cancer survivors and had agreed to be contacted for future research.

We have added this detail to page 5, lines 82-86.

Results

8. L119 Fifty-two of 336 survey participants were eligible for this study - was it that they were eligible - i.e. you had eligibility criteria? Or that 52 agreed to take part? I'm asking as Table 1 suggests all survey respondents might be eligible. Just clarify please.

Response: We apologise that this was unclear. Please see our response on point 7.

9. L131 Title - Survivor-centred barriers to provision. again for clarity might you preface with "HP's perceptions of" survivors centred. just to make it really explicit this was primary research with HP's not cancer survivors.

Response: We have prefaced this as suggested (page 7, line 133).

10. L138 sub-optimal (hyphen)

Response: This has now been corrected.

11. L155 Title - consider revising to - HP's perception of cancer survivors ability to perform health behaviours. It also seems the first quote is talking about appropriateness of advice that might be given

Response: We have revised this as suggested (page 8, line 164).

12. L167 mood and distress sentence and quote might sit better in the previous 1.1 section - patient barriers

Response: We have moved this as suggested.

13. L223 I think there are some important distinctions to be made between giving general lifestyle advice with the intention of improving quality of life, general health and advice that is specifically focussed upon reducing risk of disease recurrence.

Response: This point has now been clarified in section 2.1 highlighting that the advice tended to focus on either general health or controlling side effects and improving recovery. Only some HPs advised survivors that healthy lifestyle might improve disease outcomes.

14. Can you clarify the point you make about the evidence for physical activity being stronger - was that in terms of general health advice or in terms of reducing further cancer recurrence?

Response: This has now been clarified to note that the point refers to evidence for physical activity on cancer recurrence (page 10, lines 252-254).

15. Do you have any further data that gives any insight as to for what purpose specifically HP's give advice - it is generic healthy lifestyle advice or do they say it's for specific reducing further disease risk? Is there any consensus you can glean from your data? I appreciate you have stated that HP's do not know the evidence base sufficiently well to make clear recommendations - but is that what they want to be able to do?

Response: This point has now been clarified in section 2.1 highlighting that the advice tended to focus on either general health or controlling side effects and improving recovery. Only some HPs advised survivors that healthy lifestyle might improve disease outcomes. An additional quote has been added.

16. L256 Clinicians or HP's? - choice of term Please can you clarify sentence - couldn't quantify effect of lifestyle changes post-treatment - in terms of survival, recovery, wellbeing?

Response: This has now been addressed.

17. L290 They - can you say HP's - to be explicit who you are referring to

Response: This has now been corrected to note that it refers to HPs (page 12, line 316).

18. L342 Clinicians or HP's

Response: This has now been corrected to HPs.

19. L346 They felt all HP's should provide lifestyle advice - who is they?

Response: This has now been corrected to "HPs felt that all of them should provide lifestyle advice" (page 14, lines 371-372).

20. L352 Those in favour - again just clarify who you are referring to

Response: Apologies, we were referring to HPs. This has now been clarified.

Discussion

21. L411 achievable goals framed as part of their treatment regimen - I agree this sounds like a good plan - do you think the strength of the presented data/quotes is strong enough to infer this is what HP's wanted / were meaning?- that lifestyle advice and changes be viewed as treatment regimen? I think you would need to substantiate with more quotes to make this bold statement. When you read the quotes it suggests that HP's don't have the knowledge re evidence, have perceptions of patient's behaviours and fear losing that connection - would you just consider how you might be able to substantiate that point further. Maybe the term "care" is more appropriate - treatment has a medical implication?

Response: We have included an additional quote to support this statement. As we mentioned in the text, this was something mentioned by only some HPs. We have edited the discussion to emphasise that this was a suggestion, but retained the use of 'treatment' as this was the term used by the participant themselves.

22. L440 Reference 11 - you make a valid point about the cancer prevention recommendations - could you add / refer to the Rock (2012) guidelines specifically for cancer survivors.

Response: Thank you for this suggestion, this has now been added.

23. General Clearly written, good supporting up to date referencing, interesting points that affirm there is still a lot of work to do to incorporate effective lifestyle advice for cancer survivors into routine cancer care.

Response: Thank you for your comment.

We would like to thank the editors and reviewers for taking the time to review our manuscript and giving us thoughtful feedback. We have tried to address their comments and hope that this has improved our manuscript. We have also updated the page numbers in the COREQ checklist to reflect these changes. Please see our responses below, and the corrections and additions in the revised manuscripts.

Reviewer 1

1. The aim could benefit from being restructured.

Response: Thank you, we have restructured our aim to read: 'The aim of this qualitative study was to explore in greater depth HPs' views on the provision of lifestyle advice to patients with a cancer diagnosis' (page 4, lines 78-79).

2. It would be helpful to include justification for why HPs of breast, prostate, or colorectal were selected. This has been provided in line 472, but should be clarified in the first instance
Response: As suggested, we have moved this justification to the beginning of the methodology section (page 5, lines 82-84).

3. Clarify eligibility criteria (eg on quote refers to end of life care which is quite different).
Response: We apologise that this was unclear and agree that end of life care is quite different. We asked HPs about the lifestyle advice they provided to survivors focusing on the post-treatment period. However, HPs did mention that they would find it inappropriate to provide advice at the end of life stage. Therefore, we thought that it would be useful to report this. However, we agree it is confusing and have removed this statement. We have also added to our methodology that the interviews were about the provision of advice to patients who had completed treatment (page 5, line 98-99).

4. Justify content analysis as chosen methodology
Response: We have now added the following detail to our methodology 'Content analysis was chosen as the main aim of the study to describe HPs views as opposed to developing theory or exploring the lived experience.' We wished to employ a relatively low level of interpretation and so content analysis was appropriate as opposed to those approaches that require a higher level of interpretive complexity (page 5, lines 108-109).

5. Barriers to supporting cancer survivors to make lifestyle changes seems to have limited attention in the interview schedule – was this reflected in the transcripts?
Response: We are unclear about this point, as topic 3 of the interview schedule covers barriers to provision of lifestyle advice with 9 questions. Every participant discussed the barriers in detail and this was reflected in the transcripts and results.

6. Could you elaborate on why field notes weren't taken?
Response: Field notes were not taken as all but one interview took place over the phone, meaning information could not be collected on setting, appearance or nonverbal behaviours of participants. We have added this information to our methodology (page 5, line 103-105).

7. Please clarify what is meant by 'comprehensive examination' of HPs views?
Response: We apologise that this was unclear. We meant that we covered the key modifiable health behaviours. This has now been corrected.

8. In section 1.2 there is a quote that refers to frailty but not for tolerability of a high-fibre diet. Can a quote be provided for this as well?
Response: We have now provided a quote for this as well (page 8, lines 173-176).

9. Line 188-191: statement doesn't feel justified by the quotes provided
Response: Further quotes have been added which we hope better justify this statement (page 9, lines 194-202).

10. Line 340: Typo
Response: Thank you, the typo has been corrected.

11. Discussion – avoid statements that present as definitive conclusions
Response: Apologies, we have removed these statements and presented them more cautiously.

12. Line 475-477: The conclusion needs to reflect the results. For example, HPs described barriers to provision of lifestyle advice, such as only having capacity to provide general advice rather than giving tailored information or providing advice that will elicit a survivors to make behaviour change. Should

line 476 read "of those who provided advice", as currently it gives the impression that respondents provided advice

Response: Thank you, we have now reworded this sentence in line with this suggestion (page 18, line 505).

13. This year the WCRF also released CUP reports for both breast and colorectal cancer. These could also be included in your references

Response: Thank you for these suggested references. While they are of interest, we have chosen not to include these given their focus on prevention. We feel the more generic WCRF guidelines, while older, are more relevant as they specifically address survivorship.

Reviewer 2

1. L22 semistructured - change to semi-structured

Response: Thank you, this has now been corrected.

2. L28 2) Clinician-centred barriers - are you using the term clinician interchangeably with HP's -? Are you specifying a type of professional by using the term clinician? can you check through-out if you want to keep in clinician or just use HP.

Response: This has now been corrected and "HPs" is used throughout the manuscript.

3. L29 "Results suggested that survivors current". could you pre-face with "HP's perceptions of survivors." just to make it clear you are reporting from the HP's point of view on survivors behaviours.

Response: Thank you, this has now been corrected (page 2, line 28-29).

4. L44 "on all major lifestyle topics" - consider revising to on the key modifiable lifestyle behaviours

Response: We have revised this as suggested (page 3, line 44).

5. L72 Consider revising to - previously we conducted a survey with HP's suggesting - to make it explicit that work was already undertaken.

Response: We have revised this as suggested (page 4, line 70).

6. L77 "belief that they would blame the patient" - not easy to know what you mean by this - perhaps remove.

Response: As suggested, we have removed this statement.

7. Methods Could you clarify whom was invited to take part in the interview study - was it all previous survey respondents received an email invitation? see next point

Response: We apologise that this was unclear. Eligible participants were UK-based HPs (including surgeons, physicians, nurses and allied HPs) caring for patients diagnosed with the most common lifestyle-related cancers (breast, prostate, or colorectal cancer), who had provided contact details following their participation in an online survey on lifestyle guidelines for cancer survivors and had agreed to be contacted for future research.

We have added this detail to page 5, lines 82-86.

Results

8. L119 Fifty-two of 336 survey participants were eligible for this study - was it that they were eligible - i.e. you had eligibility criteria? Or that 52 agreed to take part? I'm asking as Table 1 suggests all survey respondents might be eligible. Just clarify please.

Response: We apologise that this was unclear. Please see our response on point 7.

9. L131 Title - Survivor-centred barriers to provision. again for clarity might you preface with "HP's perceptions of" survivors centred. just to make it really explicit this was primary research with HP's not cancer survivors.

Response: We have prefaced this as suggested (page 7, line 133).

10. L138 sub-optimal (hyphen)

Response: This has now been corrected.

11. L155 Title - consider revising to - HP's perception of cancer survivors ability to perform health behaviours. It also seems the first quote is talking about appropriateness of advice that might be given

Response: We have revised this as suggested (page 8, line 164).

12. L167 mood and distress sentence and quote might sit better in the previous 1.1 section - patient barriers

Response: We have moved this as suggested.

13. L223 I think there are some important distinctions to be made between giving general lifestyle advice with the intention of improving quality of life, general health and advice that is specifically focussed upon reducing risk of disease recurrence.

Response: This point has now been clarified in section 2.1 highlighting that the advice tended to focus on either general health or controlling side effects and improving recovery. Only some HPs advised survivors that healthy lifestyle might improve disease outcomes.

14. Can you clarify the point you make about the evidence for physical activity being stronger - was that in terms of general health advice or in terms of reducing further cancer recurrence?

Response: This has now been clarified to note that the point refers to evidence for physical activity on cancer recurrence (page 10, lines 252-254).

15. Do you have any further data that gives any insight as to for what purpose specifically HP's give advice - it is generic healthy lifestyle advice or do they say it's for specific reducing further disease risk? Is there any consensus you can glean from your data? I appreciate you have stated that HP's do not know the evidence base sufficiently well to make clear recommendations - but is that what they want to be able to do?

Response: This point has now been clarified in section 2.1 highlighting that the advice tended to focus on either general health or controlling side effects and improving recovery. Only some HPs advised survivors that healthy lifestyle might improve disease outcomes. An additional quote has been added.

16. L256 Clinicians or HP's? - choice of term Please can you clarify sentence - couldn't quantify effect of lifestyle changes post-treatment - in terms of survival, recovery, wellbeing?

Response: This has now been addressed.

17. L290 They - can you say HP's - to be explicit who you are referring to

Response: This has now been corrected to note that it refers to HPs (page 12, line 316).

18. L342 Clinicians or HP's

Response: This has now been corrected to HPs.

19. L346 They felt all HP's should provide lifestyle advice - who is they?

Response: This has now been corrected to "HPs felt that all of them should provide lifestyle advice" (page 14, lines 371-372).

20. L352 Those in favour - again just clarify who you are referring to

Response: Apologies, we were referring to HPs. This has now been clarified.

Discussion

21. L411 achievable goals framed as part of their treatment regimen - I agree this sounds like a good plan - do you think the strength of the presented data/quotes is strong enough to infer this is what HP's wanted / were meaning?- that lifestyle advice and changes be viewed as treatment regimen? I think you would need to substantiate with more quotes to make this bold statement. When you read the quotes it suggests that HP's don't have the knowledge re evidence, have perceptions of patient's behaviours and fear losing that connection - would you just consider how you might be able to substantiate that point further. Maybe the term "care" is more appropriate - treatment has a medical implication?

Response: We have included an additional quote to support this statement. As we mentioned in the text, this was something mentioned by only some HPs. We have edited the discussion to emphasise that this was a suggestion, but retained the use of 'treatment' as this was the term used by the participant themselves.

22. L440 Reference 11 - you make a valid point about the cancer prevention recommendations - could you add / refer to the Rock (2012) guidelines specifically for cancer survivors.

Response: Thank you for this suggestion, this has now been added.

23. General Clearly written, good supporting up to date referencing, interesting points that affirm there is still a lot of work to do to incorporate effective lifestyle advice for cancer survivors into routine cancer care.

Response: Thank you for your comment.

VERSION 2 – REVIEW

REVIEWER	Claire Foster and Amy Din University of Southampton, UK
REVIEW RETURNED	25-Jan-2018

GENERAL COMMENTS	The authors have provided a very thorough response to the reviewers' comments and have almost addressed comments to our satisfaction. There remains one point that requires a little more explanation: 1. A list of "key modifiable Health behaviours" alongside the use of the term (eg in the introduction).
---

REVIEWER	Helen Crank The Centre for Sport and Exercise Science Academy of Sport and Physical Activity Sheffield Hallam University United Kingdom
REVIEW RETURNED	02-Feb-2018

GENERAL COMMENTS	Thanks to the authors for working through the suggested revisions. In my opinion the manuscript provides very useful insights into the challenges of delivering lifestyle advice to cancer survivors. It is clear there is still considerable work to be done to make lifestyle advice a routine feature in cancer survivorship care.
---

VERSION 2 – AUTHOR RESPONSE

We would like to thank the editors and reviewers for taking the time to review our manuscript again and giving us thoughtful feedback.

Response to point by Reviewer 1:

We have now addressed the point raised by explaining the term "key modifiable lifestyle behaviours" in the introduction and using it consistently throughout the manuscript.

Response to point by Reviewer 2:

Thank you for your comment.